# Histological and Immunohistochemical Characterization of Osteoimmunological Processes in Scaffold-Guided Bone Regeneration in an Ovine Large Segmental Defect Model

**DOI:** 10.3390/biomedicines11102781

**Published:** 2023-10-13

**Authors:** Ronja Finze, Markus Laubach, Mairim Russo Serafini, Ulrich Kneser, Flavia Medeiros Savi

**Affiliations:** 1Centre for Biomedical Technologies, School of Mechanical, Medical and Process Engineering, Queensland University of Technology, Brisbane, QLD 4059, Australia; ronja.finze@hdr.qut.edu.au (R.F.);; 2Department of Hand-, Plastic and Reconstructive Surgery, Burn Center, BG Trauma Center Ludwigshafen, University of Heidelberg, 67071 Ludwigshafen, Germany; ulrich.kneser@bgu-ludwigshafen.de; 3Australian Research Council (ARC) Training Centre for Multiscale 3D Imaging, Modelling and Manufacturing (M3D Innovation), Queensland University of Technology, Brisbane, QLD 4000, Australia; 4Department of Orthopaedics and Trauma Surgery, Musculoskeletal University Center Munich (MUM), LMU University Hospital, LMU Munich, 81377 Munich, Germany; 5Department of Pharmacy, Universidade Federal de Sergipe, Sao Cristovao 49100-000, Brazil; maiserafini@hotmail.com; 6Max Planck Queensland Center for the Materials Science of Extracellular Matrices, Queensland University of Technology, Brisbane, QLD 4059, Australia

**Keywords:** immunohistochemistry, foreign body reaction, bone defect, scaffold-guided bone regeneration, polycaprolactone, in vivo, sheep

## Abstract

Large-volume bone defect regeneration is complex and demands time to complete. Several regeneration phases with unique characteristics, including immune responses, follow, overlap, and interdepend on each other and, if successful, lead to the regeneration of the organ bone’s form and function. However, during traumatic, infectious, or neoplastic clinical cases, the intrinsic bone regeneration capacity may exceed, and surgical intervention is indicated. Scaffold-guided bone regeneration (SGBR) has recently shown efficacy in preclinical and clinical studies. To investigate different SGBR strategies over periods of up to three years, we have established a well-characterized ovine large segmental tibial bone defect model, for which we have developed and optimized immunohistochemistry (IHC) protocols. We present an overview of the immunohistochemical characterization of different experimental groups, in which all ovine segmental defects were treated with a bone grafting technique combined with an additively manufactured medical-grade polycaprolactone/tricalcium phosphate (mPCL-TCP) scaffold. The qualitative dataset was based on osteoimmunological findings gained from IHC analyses of over 350 sheep surgeries over the past two decades. Our systematic and standardized IHC protocols enabled us to gain further insight into the complex and long-drawn-out bone regeneration processes, which ultimately proved to be a critical element for successful translational research.

## 1. Introduction

Scaffold-guided bone regeneration (SGBR) has been successfully translated into clinical applications in the field of regenerative medicine to advance the treatment of bone defects when the defect zone exceeds the critical size for the self-healing capacity of the organ bone [1,2,3,4,5,6,7,8]. It is important to notice that fracture healing and scaffold guided bone regeneration are two interrelated but also inherit different time course related processes during bone regeneration, each with unique characteristics and mechanisms. Several reviews have summarized today’s knowledge and research regarding fracture healing, and we refer the reader to these publications [9,10]. Hence, in this publication, we focused on the histological and immunohistochemical SGBR, which has been recently described in detail by our group [11].

In the last two decades, our group has extensively studied mPCL and mPCL-TCP scaffolds in vitro and in vivo, with the focus on mechanical properties, degradation, geometry, surface, and fabrication optimization, as well as in vivo biocompatibility [12,13,14,15,16,17,18,19]. Based on specimen analyses from a multitude of large animal studies [12,13,16,20,21], we have been able to demonstrate that bone regeneration at the scaffold–tissue interface, as well as throughout the scaffold, relies on a comprehensive and complex network of processes involving hemostasis, the immune response, neovascularization, callus development, and ultimately remodeling into functional, highly organized bone tissue. On closer examination, the decisive regeneration of the organ bone is influenced by the initial inflammatory state of the local microenvironment, in which macrophages facilitate the release of osteogenic cytokines and, therefore, affect the formation of new tissue around and throughout implanted scaffolds, also conceptualized in the more recent literature as osteoimmunomodulation [22]. Therefore, it is imperative to investigate the entirety of the subcomponents involved to comprehend the complexity of the entire regeneration concept of SGBR, whose bone defects may take up to several years to regenerate.

The introduction of a foreign body and its inherent immune responses, particularly when implanting mPCL scaffolds, has substantial effects on the host adaptative immune system [23]. Past research has indicated that there are more than 6500 genes regulating bone regeneration, which suggests a complex biological niche in which multiple cell interactions occur to promote functional bone formation [24]. The in vivo evaluation of the adaptive host immune responses through immunohistochemistry (IHC) analysis is an invaluable component of translational SGBR research. We have demonstrated that, with ample training and experience of the interdisciplinary team in both histological and immunohistochemical analysis of segmental defect studies in sheep, a high degree of reproducibility can be achieved when validated histological and IHC protocols are developed and stringently followed [25]. 

IHC analysis has been shown to be a means for identifying cellular markers that distinguish specific phenotypes for evaluating the osteoimmunological response in SGBR, and in this context, is a condition *sine qua non* for in-depth assessment of the spatiotemporal bone regeneration processes [26]. Mainly derived from knowledge rooted in fracture healing research, and based on the extensive literature, one can conclude that there are two different pathways for bone tissue regeneration, namely direct ossification, where new lamellar bone forms immediately, and which only occurs with a given proximity, and stability of the fracture ends. However, more commonly, and seen in SGBR, tissue of the organ bone indirectly regenerates through endochondral (cartilaginous) ossification. This form of healing appears in five interrelating steps, comprising a hematoma in combination with a stable fibrin network, granulation tissue, soft callus (cartilage), and hard callus (woven bone) formation, and lastly bone remodeling [27]. However, steps of SGBR occur both in a sequential overlay and are concurrent; thus, the exact chronological allocation to a distinct ‘state of healing’ in contrast to the healing of a simple fracture is impossible. Our extensive experience from over 12 studies and more than 350 large segmental sheep surgeries has demonstrated that these aspects are critical to overall success and the ability to assess histologic and IHC findings regarding bone defect healing at different time points. These skills in preparing and evaluating histologic and immunohistological specimen reproducibly and validity are usually learned “on the job” or through the rare internship opportunities in SGBR-specialized academic institutions.

The lack of a conceptual replication to determine the probative value and functionality of specific biomarkers for evaluating novel SGBR approaches in a sheep model that reflects analogous conditions in humans renders IHC reproducibility challenging and is ultimately a barrier to clinical translation. To further understand the body’s response and action on a cellular and molecular level, our group has integrated extensive IHC analyses as a fundamental component of the histoanalytical protocol for our in vivo studies. The aim of the present work was to signify the value of IHC for the assessment of tissue regeneration through the combined assessment of immune cellular, vascular, and extracellular matrix (ECM) components, whereby the context was based on work performed on our sheep large segment bone defect model [21,25,28,29,30].

## 2. Materials and Methods

Ethical approval was obtained from the Queensland University of Technology (QUT) Animal Ethics Committee (UAEC) (Ethics approval numbers: 1000001139; 1300000453; 1600000280; and 0900000906). All animal surgeries were performed at the QUT Medical Engineering Research Facility (MERF) and were in line with the requirements set out in the Australian Code for the Care and Use of Animals for Scientific Purposes. 

### 2.1. Study Groups

To illustrate the range of immunohistochemical stains validated in our research group, we have compiled a selection of different experimental test groups for this study. The details of the experimental groups are presented in Appendix A. The biodegradable 3D-printed scaffolds (Appendix A) were fabricated via fused deposition modeling (FDM) with rectilinear filling and purchased from Osteopore International Pte Ltd. (Singapore) and composed of mPCL (80 wt%) and β-TCP (20 wt%) (Appendix A). The scaffolds were either additionally loaded with recombinant human bone morphogenic protein-7 (rhBMP-7, 2 mg) delivered in platelet-rich plasma (PRP) or implanted either in combination with Reamer–Irrigator–Aspirator (RIA) system bone graft material, an iliac crest bone graft (ICBG), or in combination with a corticoperiosteal flap (CPF). Applying the RIA system implies harvesting bone graft material from the medullary canal of the femur or tibia, and represents the standard clinical method for the bone graft material collection of large segmental defects [31]. The RIA system’s bone graft material was harvested in the same tibia as the applied segmental defect and subsequently implanted in the defect void. Details of the surgical technique of scaffold implantation in combination with the CPF are described in detail elsewhere [32].

### 2.2. Surgical Protocols

In all our experiments, male Merino sheep (50–60 kg bodyweight, age ≥ 6 years) were utilized. The key stages in the surgical approach for the sheep tibial segmental defect were previously published in detail [25]. Briefly, the skin and soft tissue in the medial leg were incised to facilitate exposure of the tibia while preserving the neurovascular bundle. Hohman hooks were placed to ensure that the neurovascular bundle is adequately protected, and osteotomies were performed to create the defect. The periosteum was also removed proximally and distally to the defect for a length of at least 5mm to reduce spontaneous healing. The treatment of the defect void was performed as per the study groups (Appendix A). The defect was fixed with a human-sized dynamic compression plate (DCP) containing a range from 10 to 12 holes, with the scaffold placed centrally and secured to the plate with a single Vicryl suture (Appendix A). The soft tissue and skin were then closed in layers. Postoperatively, analgesia, temporary full-leg cast application, as well as sling support in a customized sheep sling, were provided as per our validated and standardized peri-operative protocols [25]. Humane killing was performed at different time points, with the earliest time point at six hours and the latest time point of 36 months. After euthanasia, the freshly harvested (affected and unaffected) tibias were biomechanically evaluated with torsional loading after plate removal.

An important principle of histology and immunohistochemistry is to optimize the integrity of the specimens and the interface between the implant and the host tissue. A microscopic evaluation of an implantation or surgical procedure requires the utmost protection of the implant–tissue interface and surgical planes. It is important to establish a dialogue between the harvesting team and the histology specialist at the time of harvesting the tissue construct in order to train all parties involved to refine and adhere to the protocols for specimen collection. Therefore, communication within the team is relevant to note the in situ sample’s orientation, as well as to keep the ex vivo time without fixation as short as possible, as the longer a tissue is unfixed, the greater the chance for degradation of the target biomarker, especially if it is a protein and subjected to autolytic processes. For immersion fixation, a fixative volume of 15–20:1 in a fixative-to-tissue ratio is recommended, and the absolute minimum would be a fixative volume of 10:1 in tissue ratio. Therefore, following biomechanical testing, the scaffold specimens were removed and fixed in 4% paraformaldehyde for seven days before being transferred to 70% (*v*/*v*) ethanol until needed for further analysis.

### 2.3. Immunohistochemical Analysis

To assess the inflammatory response, vascularization, and ECM deposition, cellular, morphological, and microstructural details of the defect samples were visualized using paraffin-embedded specimens. A schematic overview of the immunohistochemical sectioning planes is provided in Appendix A. To allow histological assessments of the decalcified paraffin-embedded samples via IHC, fixed samples were, in accordance with previously defined and standardized multiple parts, cut using an EXAKT 310 diamond band saw (EXAKT Apparatebau GmbH & Co.KG, Norderstedt, Germany) [20,25]. Decalcification was performed in 10% EDTA within the time range from eight to 10 weeks at 37 °C using a rapid decalcifier (KOS Rapid microwave lab station, ABACUS, Brisbane, Australia). Next, the samples were serially dehydrated in ethanol in an automated Excelsior ES tissue processor (Excelsior ES, Thermo Scientific, Franklin, MA, USA) and embedded in in molten paraffin wax at 60 °C. Sections at a 5 μm thickness from the paraffin samples were cut using standard rotary microtomes (Leica Biosystems, Nussloch, Germany) and disposable blades and collected onto polylysine-coated microscope slides and dried at 60 °C for 16 h. Aside from performing standard hematoxylin and eosin (H&E) staining using a Leica Autostainer XL (Leica Biosystems, Nussloch, Germany) and histological staining, a subset of slides were used for immunohistochemistry, and were performed according to validated and standardized protocols previously established by our group [20,25]. The primary antibodies used for this study, which are specific to the inflammatory response, vascularization, and ECM (osteogenesis) markers, are listed in Appendix A. The primary antibodies represented in this study for the inflammatory response are inducible nitric oxide synthase (iNOS), mannose receptor (MR), Arginase-1 (ARG-1), cluster of differentiation 68 (CD68), transcription factor interferon regulatory factor (IRF5), cluster of differentiation 3 (CD3), and cluster of differentiation 45 (CD45), and for the vascular endothelial cells (VECs) are vascular endothelial growth factor (VEGF), cluster of differentiation 31 (CD31), von Willebrand factor (vWF), alpha smooth muscle actin (α-SMA), angiopoietin-1 (ANG1), Noggin, and Notch 1, and for the ECM (osteogenesis) include collagen type I (COL I), collagen type II (COL II), bone morphogenetic protein 2 (BMP-2), osteoprotegerin (OPG), alkaline phosphatase (ALP), sclerostin (SCL), osteomodulin (OMD), osteonectin (ON), osteopontin (OPN), and osteocalcin (OC). Additionally, α-SMA was also used as a marker for scaffold encapsulation.

## 3. Results

A dataset collection of SGBR-relevant protein expression of the investigated antibody markers in five experimental groups, with follow-ups ranging from six hours to 36 months (a total of nine different time points), and their specifically assessed cells and tissues are illustrated in Figure 1. Overall, our dataset results did not identify pronounced differences in the osteoimmunological response to the different mPCL-TCP scaffolds (mPCL-TCP or mPCL-TCP-CaP), nor when comparing different time points and experimental groups. COL I deposition was observed in all experimental groups and at all different time points throughout the open and fully interconnected porous architecture of the scaffolds (Figure 1 COL I, A–M). Overlapping stages of woven bone and lamellar bone formation were identified through a stronger and lighter intensity of the COL I^+^ stain, respectively (Figure 1 COL I, G). Although the newly formed bone matrix showed some islands of remnant mineralized cartilage matrix (Figure 1, COL II, A–M, red arrowheads) at proximity to the mPCL-TCP struts, no fibrous tissue formation was observed enclosing the mPCL-TCP struts. OC^+^ expression was observed at the osteoblasts (Figure 1 OC, A, B, E, G and L, green arrow heads) and at the osteoid matrix as early as six hours, with apparent secondary osteon formation with a strong OC^+^ stain at cement lines at 12, 15, 21 months, and 36 months (Figure 1 OC, D, F, I, K, and M, black arrow heads). Neovascularization could be seen throughout the newly formed tissue and proximate to the mPCL-TCP scaffold struts (Figure 1, vWF A–M, green arrows). This newly stablishing vasculature, especially at the scaffold strut interfaces, as well as the osteoclast cells present in cutting cones crossing the newly bone formed at the defect site (Fig. 1 CD68 B-M, red arrows), and remnants of hypertrophic cartilage (Fig. 1 CD68 A) were continuously stained by CD68^+^ (M1 and M2 macrophage marker) throughout the experimental time points. The M1 macrophage marker iNOS was particularly observed at remnants of mineralized cartilage (Figure 1 iNOS C, F, H, and L, double black arrowheads) preceding vascularization, and at blood vessels around the mPCL-TCP scaffold struts (Figure 1 iNOS A, B, D, E, G, I, and M, black arrows); however, its detectability decreased overtime, only remaining detectable at blood vessels. The M2 macrophage marker MR was strongly expressed throughout all time points and experimental groups (Figure 1 MR A–M). This reactivity was located at the vessels (Figure 1 MR F, I, J–M, double red arrowheads) and osteoclasts surrounding the outer surface of scaffold struts and cutting cones (Figure 1 MR D, J, and K, red arrowheads), at fragments of bone graft and host bone (Figure 1 MR A, B, and M red arrows), and at remnants of mineralized cartilage (Figure 1 MR C and H, double black arrowheads). No increase in the M1/M2 ratio indicating a chronic foreign body reaction (FBR) was observed, further corroborated with lack of α-SMA^+^ marker expression. Negative controls are provided in Appendix A. Comprehensive and further analyses of inflammatory markers, including ARG1^+^, IRF5^+^, CD3^+^, and CD45^+^, VEC markers, such as VEGF^+^, CD31^+^, α-SMA^+^, and ANG1^+^, and ECM osteogenic markers, comprising BMP2^+^, OPG^+^, ALP^+^, SCL^+^, OMD^+^, ON^+^, OPN^+^, OC^+^, Noggin^+^, and Notch 1^+^, have also been studied and discussed in the following sections.

### 3.1. The Inflammatory Response (Figure 2)

One of the first macrophagic responses tested when evaluating SGBR upon mPCL-TCP implantation is staining samples for CD68^+^, a marker of both the M1 and M2 macrophage subtypes. The immunoreactivity of CD68^+^ was constant and clearly identified at the endothelial tissue around the mPCL-TCP struts and within the cutting cones throughout the newly formed and remodeled bone tissue (Figure 2 CD68, A). Local and phenotypic differences of these macrophagic immunoreactions were observed and distinguishable. Classically activated M1 macrophages increase their phagocytic activity and produce pro-inflammatory cytokines, including the expression of iNOS^+^ and IRF5^+^. The expression of iNOS^+^ and IRF5^+^ was mainly localized at the sites of neovascularization, especially at the interface to the mPCL-TCP scaffold struts, where newly formed vessels were formed, and at remnant areas of mineralized cartilage matrix (Figure 2 iNOS and IRF5, A). Positive controls also denoted iNOS^+^ and IRF5^+^ at vascular tissues of the liver and spleen (Figure 2 iNOS B and IRF5 B, respectively), and within vessels of the medullary canal of the contralateral tibia (Figure not shown). Conversely, M2 macrophages, including the MR marker, were strongly expressed at areas of bone resorption, and at the cells lining the surface of the newly formed bone around the mPCL-TCP scaffold struts (Figure 2 MR, A). Although less pronounced, ARG-1^+^, which is also an M2 macrophage marker and known to promote fibroblast proliferation and block iNOS activity, was positively stained at the same previously described areas (Figure 2 ARG-1, A). We further tested samples against hematopoietic cells, including the CD45 and CD3 markers. CD45^+^ expression was specifically restricted to the osteoclasts mediating bone remodeling at the cutting cones throughout the newly formed bone (Figure 2 CD45, A). Although the positive control was immunoreactive for CD3^+^, the tested mPCL-TCP samples did not show reactivity (Figure 2 CD3, A). It was observed that iNOS^+^ expression decreased, and MR expression increased overtime around the mPCL-TCP scaffold struts. However, iNOS expression was continuous throughout the early and late time points at the remnants of mineralized cartilage and endothelial vascular cells, respectively. All samples were tested with positive (Figure 2 Bs) and negative (Appendix A) controls.

**Figure 2 biomedicines-11-02781-f002:**
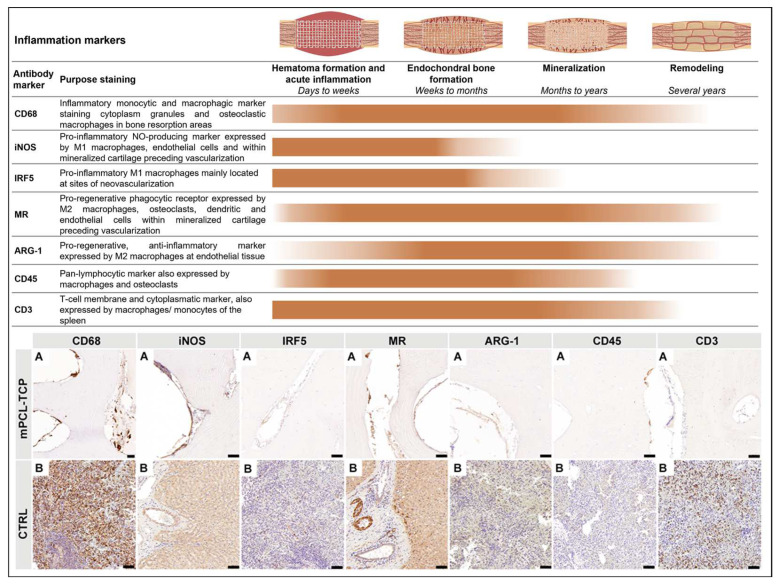
The first elicited cellular reaction upon biomaterial implantation is the initial inflammatory response, caused by the release of M1-subtype macrophages (iNOS and IRF5), exerting angiogenic functions among ECM clearing. These macrophages mainly populate the outer surface of the mPCL-TCP scaffolds and are present at sites of neovascularization, especially at the interface between the scaffold struts and the newly formed bone. As regeneration progresses, the release of anti-inflammatory cytokines, which includes the expression of multiple M2 macrophage subtypes (MR and ARG-1), is more evident and present at the outer surface of the mPCL-TCP scaffolds. (**A**) experimental samples; (**B**) positive controls; NO = nitric oxide; nitric oxide synthase (iNOS); mannose receptor (MR); Arginase-1 (ARG-1); cluster of differentiation 68 (CD68); interferon regulatory factor 5 (IRF5); cluster of differentiation 3 (CD3); and cluster of differentiation 45 (CD45). Scale bars: 50 µm. Image partially created with BioRender.com.

### 3.2. Vascularization (Figure 3)

The initial invasion of blood vessels within the cartilage matrix as early as seven days into the process of bone regeneration was observed. Several bud-shaped protrusions with slender projections connecting one bud to another and emerging from blood vessels and resembling sprouting angiogenesis spanned the entire process of SGBR. Intriguingly, thin blood vessel walls arising from the proximal site of the host bone invaginating into the defect site, and blood vessel splitting into two blood vessels, with yet another two bud-shaped protrusions branching from each end were observed. VEGF^+^ and ANG1^+^ angiogenic immunoreactivity responses were strongly observed at the endothelial tissue around the mPCL-TCP struts, and within cutting cones and haversian canals of the newly built bone tissue (Figure 3 VEGF, A and ANG1, A). Particularly, VEGF^+^ immunoreactivity was also detected at regions of cartilage undergoing endochondral ossification, and at regions of woven bone proceeding the osteochondral matrix and preceding vascular invasion at the vicinities of the mPCL struts (Figure 3 VEGF, A). High levels of the cell adhesion molecule CD31^+^ were strongly observed within the sprouting walls of blood vessels and bud-shaped sprouting blood vessels of newly formed bone tissue around the mPCL-TCP struts and at mineralized cartilage islands preceding vascularization invasion (Figure 3 CD31, A). Notably, the expression of CD31^+^, which is associated with type H blood vessels [33,34], was also prominent within the osteochondral matrix, particularly in areas where VEGF expression was more concentrated. The VEGF marker was not expressed in (positive) controls of the contralateral sheep tibia; however, ANG1 was strongly positive at the blood vessels, osteocytes, and at the endosteal tissue lining the marrow cavity (Figure 3 VEGF, B and ANG1, B). Vascular network maturation was indicated through strong staining for vWF at previously VEGF- and ANG1-stained areas, except for the mineralized cartilage areas. Contralateral sheep tibiae (positive controls) were strongly immunoreactive for CD31^+^ at smaller and larger caliber blood vessels, whereas vWF^+^ was particularly strongly immunoreactive at elongated, matured blood vessels (Figure 3 vWF, B). α-SMA^+^ reactivity was not observed adjacent to the mPCL-TCP scaffold struts, but only seen within the walls of larger caliber blood vessels in the experimental groups and contralateral sheep tibiae positive controls (Figure 3 α-SMA, A, B). Noggin^+^ expression was sparce, with limited expression in the endothelial tissue around the mPCL-TCP scaffold struts (Figure 3 Noggin, A). Conversely, Notch 1^+^ expression was detected in areas undergoing delayed endochondral ossification at the sprouting blood vessels, at the periosteum layer, as well as within the endothelial tissue of cutting cones and the endothelial tissue around the mPCL struts (Figure 3 Notch 1). Contralateral sheep tibiae (positive controls) were faintly immunoreactive for Noggin^+^ and Notch 1^+^. All samples were tested with positive (Figure 3 Bs) and negative (Appendix A) controls.

**Figure 3 biomedicines-11-02781-f003:**
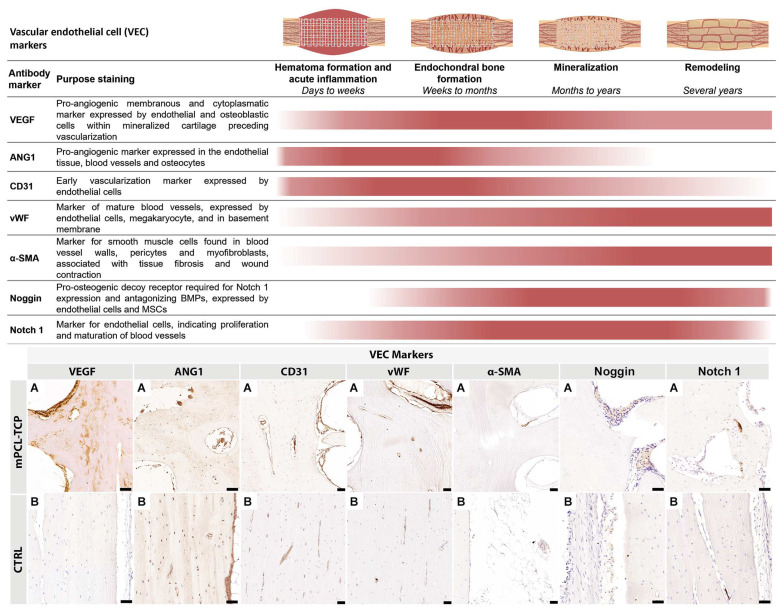
Vessel recruitment is largely coordinated through the release of vascular endothelial growth factor (VEGF) and angiopoietin 1 (ANG1) through monocytes present at disrupted vessels of the injured site, as well as through the pro-inflammatory environment initially created during hematoma formation. These newly formed vessels express high levels of platelet endothelial cell adhesion molecule (PECAM 1, also known as cluster of differentiation 31 (CD31)). This timely organized pro-inflammatory environment is precisely orchestrated to shift towards a pro-regenerative environment, which, in turn, will lead to the re-establishment and remodeling of the vascular network of the defect area. Mature vessels can be detected through the expression of von Willebrand factor (vWF) within the newly formed bone tissue and around the interconnected porous architecture of the mPCL-TCP scaffolds. Anti-smooth muscle actin (α-SMA) is specially expressed in myofibroblasts, which play a substantial role in scaffold encapsulation. However, it is also expressed at the endothelial walls of blood vessels and can therefore be used as a marker for vascularization. α-SMA expression, in our studies, was only detected at vessels formed around the mPCL-TCP scaffold struts and at larger vessels at the periosteal region of the newly formed bone tissue. In line with the literature [33,34], we observed that Noggin and Notch 1 play a regulatory role in angiogenesis and stain the endothelial walls of blood vessels. (**A**) experimental samples; (**B**) positive controls; BMP = bone morphogenic protein; MSC = mesenchymal stem cell; vascular endothelial growth factor (VEGF); angiopoietin (ANG1); cluster of differentiation 31 (CD31); von Willebrand factor (vWF); and anti-smooth muscle actin (α-SMA). Scale bars: 50 µm. Image partially created with BioRender.com.

### 3.3. The ECM (Figure 4)

The COL I marker was expressed in a random orientation within the scaffold macropores, as well as in a parallel-arranged manner on the scaffold struts (Figure 4 COL I, A). The COL II^+^ stain was particularly observed at the interface of the newly formed bone and host bone, as well as around the mPCL-TCP scaffold struts (Figure 4 COL II, A). Reactivity of the BMP2 marker was seen at the cells lining the outer surface of the mPCL-TCP scaffold struts, as well as at bone-lining osteoblast cells within the bone tissue (Figure 4 BMP2, A). Reactivity of the ALP marker was mainly observed at the osteoid matrix within the cutting cones, at the peripheral cortex region near the periosteum, as well as in proximity to the scaffold struts (Figure 4 ALP, A). OM marker reactivity was found at the endothelial walls of blood vessels throughout the osteoid matrix in between and surrounding the scaffold struts. Further, OM was expressed on the endosteal cells lining the cortical bone and at areas of mineralized tissues (Figure 4 OM, A). Upon a closer observation of our tested samples, ON^+^ was specifically detected at osteoblast cells within the bone matrix, as well as at the cells lining the outer surface of the mPCL-TCP scaffold struts (Figure 4 ON, A). Further, the ON marker was also present at osteocytes of the positive control samples (Figure 4 ON, B). OPN^+^ and OC^+^ were observed at areas of randomly oriented COL I-stained areas; however, where the fibers were aligned in a lamellar configuration appearing as osteons, the reactivities of OPN^+^ and OC^+^ were mainly restricted to the osteocyte cells and late-differentiated osteoblasts, respectively (Figure 4 OPN, A and OC, A). Further, OC^+^ was clearly evident at cement lines separating osteons. Positive controls also stained OPN^+^ at osteocytes throughout the sample and the endosteal tissue lining the marrow cavity (Figure 4 OPN, B). Corroborating these results, the SCL marker was positive throughout the osteoid under the early time points, and especially during the late time points, in the experimental and control samples (Figure 4 SCL, A and B). The OPG marker was strongly expressed by bone-lining cells located at the outer surface of the mPCL-TCP scaffold struts, and within the osteoid matrix, as well as by osteoclasts resorbing mineralized cartilage matrix areas (Figure 4 OPG, A). The detection of these OPG markers was absent within the newly formed tissue in the central area of the tubular scaffolds. The expression of these OPG markers at the osteocytes were observed at the host bone and positive control (contralateral tibia) (Figure 4 OPG, A and B). All samples were tested with positive (Figure 4 Bs) and negative (Appendix A) controls.

**Figure 4 biomedicines-11-02781-f004:**
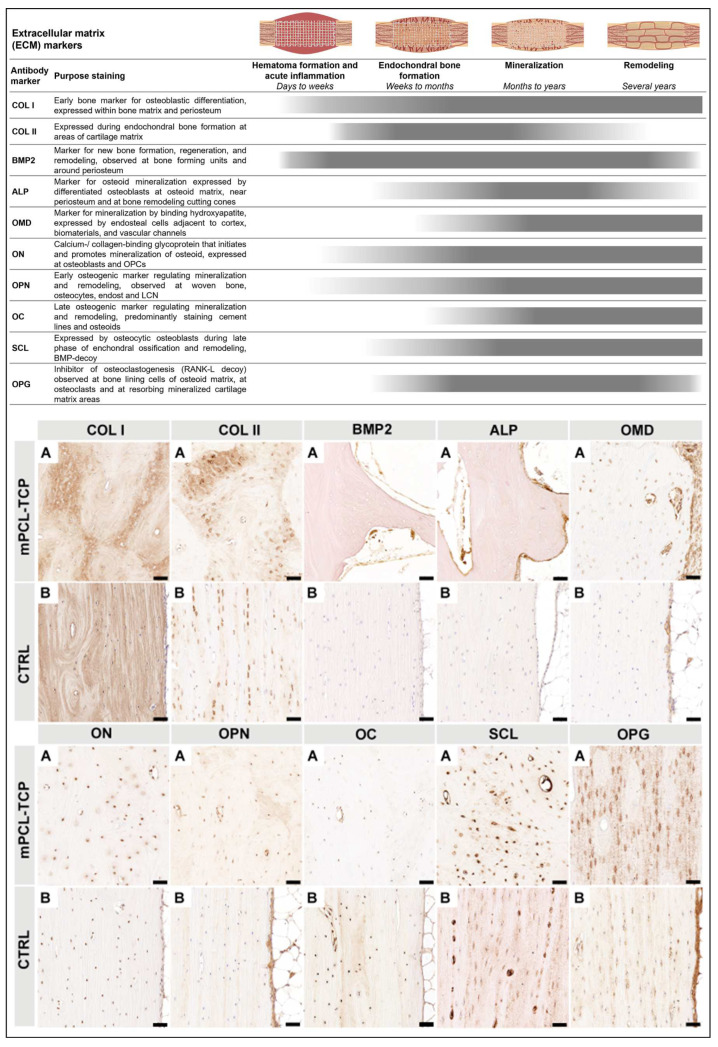
The first mineralized matrix produced during primary bone formation is resorbed by osteoclasts. As the cartilage is resorbed, secondary bone formation takes place. These two overlapping processes can clearly be observed through collagen type I deposition, depicted by COL I^+^ staining throughout the newly formed bone, as well as by COL II^+^ staining at isolated areas of mineralized cartilage, especially around the scaffold struts and at the interface between the new bone and host bone tissue. As bone regeneration continues, bone morphogenic protein (BMP) expression, observed at osteoclasts, as well as at the endothelial tissue lining the outer surface of scaffolds, stimulates osteoblast recruitment through ALP and OC signaling. ALP, an intermediate marker of osteoblast differentiation indicating bone formation and turnover (highly produced by osteoblasts and mainly seen in areas of osteoid deposition), plays an important part in local bone mineralization. Up-regulation of early osteogenic activity through OC and ON signaling follows, and secondary osteon organization and mineralization takes place. ON, a glycoprotein synthetized by osteoblasts that binds calcium, also considered an intermediate marker, then initiates mineralization and promotes mineral crystal formation. ON is positively depicted throughout the newly formed bone, bordering active osteoblasts, osteocytes, osteoid and marrow progenitor cells. Correspondingly, OC expression is also evident in osteoblasts, osteoclasts, osteocytes, at cement lines, as well as hypertrophic chondrocytes. Osteocytes synthesize SCL, a protein which mediates bone-forming osteoblast-, bone-resorbing osteoclast-, and osteocyte-cross-signaling, as well as antagonizes the activity of BMPs. Only osteocytes express SCL, which, in turn, acts in a paracrine fashion to inhibit bone formation. High levels of immunodetectable SCL are seen in osteoblasts that have reached the osteocytic stage, especially within the samples with late time points and at the bone cortex of the contralateral control samples. OPG is an early osteogenic marker, which has a pivotal role in inhibiting the osteoclast’s activation and proliferation and is therefore essential for bone resorption. Its expression was intensely found throughout the newly formed bone, especially in the samples with late time points and at the cortex of the control samples. (**A**) experimental samples; (**B**) positive controls; OPCs = osteoprogenitor cells; LCN: lacuna canalicular network; collagen type I (Col I); collagen type II (Coll II); bone morphogenic protein (BMP); alkaline phosphatase (ALP); osteomodulin (OMD); osteonectin (ON); osteopontin (OPN); osteocalcin (OC); sclerostin (SCL); and osteoprotegerin (OPG). Scale bars: 50 µm. Image partially created with BioRender.com.

## 4. Discussion

Fracture healing and scaffold-guided bone regeneration are interconnected processes involved in bone regeneration, but they possess distinct characteristics and timelines [11]. We validated the histological and IHC protocols used in the ovine tibial segmental defect model in numerous studies as a preclinical tool to evaluate the osteoimmunological responses and processes in SGBR [25].

Disruption of local soft tissue and vascular integrity depends on the extent of surgically inflicted damage; therefore, the application of protocols (e.g., for IHC) and interpretation of outcome findings are to be animal model-specific. Hence, direct comparisons between preclinical studies are only scientifically justified for the same type of animal model applied. Our research group established the large (3 cm) and extra-large (6 cm) segmental defect models as the “signature model” in SGBR, considering that each different type of model has a distinct tissue damage pattern, which also navigates the magnitude of the host body inflammatory response, and depends on the extent and type of surgical defect.

During any surgical procedure, and especially in creating a large volume segmental defect, a hematoma forms through the disruption of surrounding blood vessels and medullary leakage, initiating several well-studied and described cascadic pathways [35]. Platelets activate the coagulation system, enabling blood clot formation. Simultaneously, damage-associated molecular patterns (DAMPs) are released from impaired cells, interacting with innate immune cells, such as dendritic cells and macrophages, that subsequently deliberate chemokines (e.g., VEGF and ANG1) for further immune activation. Concurrently, mast cells degranulate and release further mediators (e.g., VEGF and histamine), promoting cellular attraction through increasing vascular permeability and enhancing neovascularization. Thus, the hematoma can be understood as a temporary matrix that forms and provides initial stability, as well as an acidic, hypoxic, and sodium- and potassium-rich environment, consisting of a fibronectin matrix, immune cells, such as monocytes, macrophages, granulocytes (neutrophil and eosinophil), and lymphocytes, as well as platelets, red blood cells, and mast cells [36]. Hence, it is apparent that the host’s metabolic changes induced through traumatic stress during surgical procedures significantly affect the healing and regeneration processes [37]. However, in the contemporary literature, FBR is primarily discussed independently of the osteoimmunological processes that occur during bone healing.

Independent of the SGBR treatments and time points employed, we conclude, from our studies’ results, that there is a clear overlap between several processes, and, undoubtably, that osteoimmunological responses to the SGBR approaches is a combination of the immunological response firstly caused by the injury site and surgical mode of implantation, and secondly by the host response to the implant material properties. The stereotypical concepts of regeneration processes postulate that successful bone regeneration is dependent on a timely orchestrated initial osteoimmunological response, which we believe does not resolve, but evolves through continuously re-programming the widely described interrelating processes of bone regeneration in response to a changing environment. Yet, these well-described processes lack an adequate characterization of the osteoimmunological responses during SGBR, including macrophage polarization in vivo, which inherently and consequently interact with biomaterials, and are still part of the adaptative osteoimmunological response of bone physiological repair. Addressing and elucidating this deficiency in characterizing the re-programming of multiple cellular interactions during SGBR emphasizes the complexity of the regulation of the osteoimmune environment, which is inherently associated with the classical (pro-inflammatory) and alternative (pro-regenerative) activations as bone restitution progresses. Therefore, these osteoimmunological responses cannot be isolated and only assigned to the initial early stages of bone fracture and hematoma formation, but are to be contemplated in their entirety over the proceeding phases of bone regeneration to adequately interpret histological and IHC datasets to dissect the mechanisms in SGBR. Based on this knowledge, we have established and optimized our protocol for histological and IHC staining, investigating the different stages of regeneration through the visualization of cellular proteins that are of immunological-, vascular-, and matrix-derived origins (Figure 1).

Host immune and inflammatory responses are traditionally associated with the defense process against harmful microorganisms; however, when discussing “inflammation” in the context of SGBR, it is important to destigmatize this term [38]. This means that inflammation does not necessarily lead to chronic inflammation and ultimately implant failure [39]. On the contrary, it contributes to eubiosis [40], and to a scaffold–host equilibrium, in which successful tissue and osteointegration is characterized by a quasi-physiological immune/inflammatory response, which is critical for peri-implant wound healing, and thus allows chronic immune surveillance to maintain tissue homeostasis [38]. However, we do not deny that it is also known that the osteoimmunological response to implants depends on the biomaterial it is made of [41]. Rather than directly comparing study groups and time points, we subscriptively examined the osteoimmunological response to the biodegradable implants in depth. Consistent with successful osteointegration, we observed CD68^+^ staining, a marker for both M1 and M2 macrophages, on the 3D-printed mPCL-TCP scaffold strut surface under the early and late time points in all experimental groups (Figure 1 and Figure 2). Thus, considering the importance of protein adsorption for subsequent macrophage attachment [36,42], we can state that the physicochemical properties of these mPCL-TCP scaffolds promote adsorption of the critical molecule(s) from wound fluids generated as a result of implantation surgery and, thereby, initiate the required osteoimmunological processes for subsequent successful tissue integration.

Moreover, IHC staining of sections with specific markers for M1 and M2 activation, namely iNOS and MR, respectively, showed labeling of the surface-adherent cells and at new blood vessels formed at the outer surface of the tested scaffold struts, at both earlier and later time points (Figure 1 and Figure 2). In the early time point study groups, we observed an initial predominance of M1 macrophages at remnants of mineralized cartilage and blood vessels, and early prevalence of M2 macrophages at fragments of bone graft, host tissue, and vascular endothelial tissue surrounding the scaffold struts. At later time points, this response evolved to an M1 prevalence at blood vessels, along with a considerable predominance of M2 at the endothelial tissue, as well as cells populating the surface of the scaffold struts (Figure 1 and Figure 2). The conceptualization of an opposing phenotypic switch from M1 to M2 macrophages, implying an adequate wound healing environment, has been reported to be a foster of tissue repair and regeneration by regulating ECM formation and organization [43,44,45]. Rather than categorizing the M1 and M2 macrophage phenotypes into two functional sub-classes, which provides a framework for macrophage polarization, we observed that both M1 and M2 phenotypes co-exist, and are continuously adapting their function in response to the repairing environment of bone regeneration. In line with previous research [46], we can also conclude that the scaffold architecture used, and the biophysical properties of the mPCL-TCP biomaterial, resulted in a balanced immune response associated with a favorable pro-regenerative environment. Furthermore, iNOS, expressed by M1 macrophages, has been reported to modulate osteoclast resorption [47,48], and to be present in the early and late stages of vascularization, playing an important role in wound healing by up-regulating endothelial cell sprouting through VEGF synthesis, and late vascular remodeling through reinstation of the vascular network [49,50,51,52]. These findings strongly corroborate the early expression of iNOS at mineralized remnants of cartilage preceding vascular invasion.

A reduction in iNOS has been found to correspond with delayed and non-union bone healing in mice, which can be ascribed to the decrease in nitric oxide (NO) production through iNOS [27,53]. NO is a free radical that is produced by different cells, such as macrophages and endothelial cells, promoting bone remodeling through osteoblastic activation when highly concentrated, and osteoclastogenesis when present in lower doses [37,49]. Thus, in the studied experimental groups, the higher early expression of iNOS throughout the entire scaffold can be interpreted as successful early neovascularization, facilitating bone graft survival and new bone formation (Figure 1 and Figure 2). Furthermore, reduced iNOS marker concentrations at later time points throughout the experimental groups were indicative for bone remodeling through osteoclast activation and genesis (Figure 1 and Figure 2). Additionally, specific transcription factors have been discovered to regulate this polarization process through the up- or down-regulation of specific interleukins, for instance, upregulation of M1 macrophages via IRF-5 [54]. In contrary, M2 macrophages are characterized by their ability to produce anti-inflammatory cytokines and increased phagocytic activity, thus supporting the high expression of MR observed at the cells lining the outer surface of our mPCL scaffold struts at the later stages of bone regeneration. Mainly expressed by macrophages, scavenger receptors, among other activities, exert phagocytic functions, thus playing a pivotal role in scaffold clearing and degradation. Like M1, M2 macrophages have also been reported to take part in angiogenic processes, e.g., by secreting VEGF, and by increasing collagen and fibroblast synthesis through ARG-1. However, in the assessed samples, ARG-1 marker reactivity was neglected. Considering that ARG-1 is only expressed by one subtype of the M2 family (M2a), it is conceivable that other pro-regenerative subtypes (M2b–d) were more predominantly involved in the osteoimmunological processes, but were not captured with the antibody panel used [36,55,56,57]. In the present work, we observed significant neovascularization (Figure 1 and Figure 3), scaffold integration, and mineralization of the (bone) tissue matrix (Figure 1 and Figure 4), and therefore, based on the corroborating results compared with previous studies [58,59], we are confident that this continuous co-existing and evolving expression of the M1 and M2 macrophages, rather than the postulated stereotypical switch from M1 to M2, ensues bone regeneration, neovascularization, and vascular remodeling.

A key component of bone formation is the re-establishment of the vascular network. It is well-known that during fracture repair, interactions between endothelial cells and osteoblasts are pivotal for coordinating vessel recruitment and subsequent repair and regeneration. However, in the context of scaffold-guided bone regeneration, where the distance between the fractured sites exceeds 3 cm, and bone repair is achieved through overlapping phases of bone regeneration, the neosynthesis of blood vessels is largely compromised, and yet, equally heterogeneous and complex in response to a dynamic and changing environment.

In the initial stages of neovascularization, early signaling molecules, such as VEGF, ANG1, and CD31, are released by monocytes adhering to the endothelium of damaged vessels [60], as well as through the hypoxic environment initially created during hematoma formation, which will eventually progress to a cartilage template. Expression of these VEC markers was observed throughout all experimental groups, particularly at the early time points, as well as under the later time points in reduced expression intensity (Figure 3). This phenomenon is congruent with the description of sprouting and intussusceptive angiogenesis [61], and aligns with the findings reported by Ramasamy et al. [62], who noted that the abundance of these blood vessel protrusions increases progressively during postnatal bone growth. The formation of these features seems to correspond to a gradient of pro-angiogenic factors, particularly VEGF, even after 12 months of remodeling, in areas undergoing delayed endochondral ossification. VEGF (Figure 3, VEGF A) was strong in proximity to the scaffold struts, indicating that the 3D scaffold architecture provides a pro-angiogenic micro- and macro-porous environment for beneficial early angiogenesis. VEGF functions as a chemotactic molecule, effectively drawing endothelial cells towards the newly forming bone matrix. It also directly regulates the differentiation and activities of osteoblasts and osteoclasts, actively participating in the process of vascular and bone remodeling. These regions of actively forming bone have previously been linked to subtype H blood vessels, likely originating from the periosteal and endosteal regions of the defect site [61]. However, it is noteworthy to mention that while Notch 1 expression was substantially strong after 12 months, Noggin expression appeared to be markedly suppressed. Yet, Notch 1 was strongly detected in blood vessels at these regions of active bone formation, especially at areas undergoing delayed endochondral ossification. Notch has been proposed as a mediator of type H vessels and a promoter of angiogenesis in the context of bone [61]. Similarly, Noggin (a BMP antagonist) has been shown to be secreted and upregulated by these type H endothelial cells, contributing to the maturation of chondrocytes and facilitating bone formation [63]. Our research findings strongly suggest that Notch plays a crucial role in the maturation and remodeling of the vascular network; however, Noggin did not appear to be associated with vascular remodeling, as proposed by Ramasamy et al. (2014) [63].

Although the previous observations underscore the complexity of the coupling mechanism of angiogenesis and osteogenesis, our IHC findings of the presented work undoubtedly support that the mPCL-TCP scaffolds, in combination with different bone graft materials, promote neovascularization throughout the highly porous morphology, thereby promoting bone regeneration, in line with clinical studies [1,2,3]. Furthermore, due to the more pronounced positive vWF staining at later time points throughout all experimental groups, it can be presumed that the present blood vessels are of mature quality (Figure 1 and Figure 3).

Notably, during endochondral bone formation (as in SGBR), cartilage provides a suitable material that is less demanding of oxygen, which temporally bridges the gap until the blood supply has sufficiently developed [64]. As the vascular bed grows into the cartilaginous tissue, the first-mineralized ECM, produced during primary bone formation, is resorbed via osteoclasts and secondary bone formation continues, which subsequently is resorbed as well. Osteoblasts express BMP2, OPG, and OPN, and secrete collagen fibrils in two forms: woven and lamellar [65]. In line with these findings, new bone formation was observed in two forms of COL I deposition (Figure 4 COL I column). First, the COL I^+^ stain was observed in a random orientation, and second, it was observed in a parallel configuration, which resembles woven and lamellar bone, respectively. Overlapping stages of woven and lamellar bone formation were observed at early and late time points throughout the mPCL-TCP scaffold and on bone grafts (Figure 1 COL I column). Early bone formation was further indicated by positive BMP2 marker expression, suggesting the recruitment of osteoblasts. At later time points, mineralized areas of cartilage depicted via COL II^+^ staining were observed at the interface of the newly formed bone and host bone, as well as around the mPCL-TCP scaffold struts, indicating mineralization and early bone remodeling through endochondral ossification. Thus, based on the favorable inflammatory response elicited via the implanted scaffolds, as well as fostered early neovascularization in all study groups, the scaffold material and design facilitated in vivo tissue integration and new bone tissue formation, resulting in a sufficiently functional organ. Furthermore, osteoid matrix mineralization was up-regulated by OMD, ON, and ALP, which play important roles in osteoblast cell growth and hydroxyapatite binding capacity, thus modulating collagen fiber shape and alignment. Bone mechanical properties are inherently associated with the collagen fiber architecture, strengthened via nestled hydroxyapatite mineral particles. An appropriate mineralization (strong OMD and ON marker expression) was observed at early time points, and was more pronounced at late time points, particularly at the bone graft-scaffold strut, as well as at the bone graft-host bone interface (Figure 4). This marker expression pattern is congruous with a strong osteointegration of the scaffold. Moreover, based on the strong expression of the OC and SCL markers under the late time points, the newly formed bone throughout the scaffolds and the scaffold-host tissue interfaces point towards being of a high quality due to the osteon arrangements with uniform cement line patterns (Figure 1 and Figure 4).

In all study groups presented in this work, very limited degradation of the scaffolds was observed, even under the late time points, as indicated by the scaffold struts that appeared as voids in our IHC findings due to the dissolution of mPCL-TCP via xylene during sample processing (Figure 1, Figure 2, Figure 3 and Figure 4, red dashed lines). Similar findings have been described in a clinical study, with a mean follow-up of 248.1 ± 435.3 days using mPCL implants in 174 consecutive patients over a period of 10 years, including a variety of burr hole craniotomies and reconstruction using mPCL implants (12 mm diameter, 5 mm thickness), with porosity maintained at 70% (Osteoplug and Osteoplug-C; Osteopore International Pte Ltd., Singapore) [4]. There is increasing evidence that the biocomposite TCP enhances cellular adhesion, neutralizes by-product acidity, and accelerates scaffold degradation [66], and has been found to improve osteogenesis through macrophage stimulation [44,66,67]. Indeed, studies have shown that the incorporation of higher ceramic contents into scaffolds can not only improve the osteogenic differentiation of osteoblasts (Figure 4 ON, A) and bone healing effects in vivo [68,69,70], but also the mechanical strength, hydrophilicity, and degradability of polymer-based composites [71,72,73].

However, it is critical to recognize that scaffold degradation during bone regeneration is a “double-edged sword”, as mPCL-TCP scaffold degradation creates space for cell infiltration and ingrowth of new bone. Excessive degradation may lead to a loss of integrity, which could limit osteoblast adhesion and proliferation, as well as triggering inflammation and chronic FBR via the massive release of degradational by-products [74,75]. Of note, ARG-1 and α-SMA are also markers that are frequently used for the detection of fibroblasts and myofibroblasts, which play a substantial role in tissue fibrosis and scaffold encapsulation. In consideration of our IHC findings, ARG-1 reactivity was faintly found at the outer surface of scaffolds (Figure 2), and α-SMA expression was only detected within vessel walls and not at the scaffold–bone interface (Figure 3), indicating no fibrous encapsulation of the biomaterial in terms of an extensive FBR. While we have already initiated further studies to test the incorporation of higher ceramic contents into mPCL matrices, we are fortunate to have a reservoir of osteoimmunological protocols and findings from previous studies, including those derived from different compositions of mPCL-bioceramic composites that will allow us to conduct direct comparisons.

In our sheep tibial segmental large defect model, standard postmortem analyses included performing plain radiography, bone volume evaluation, mechanical testing, micro-computed tomography (µCT), and histology of explanted tibial defects [25]. However, the presented study focused on immunohistochemical analyses, whereas is it noteworthy that tissue explants can be processed for usage as resin or as paraffin samples for microscopic analysis. Resin sample processing does not require decalcification and, therefore, its focus is on providing in-depth information regarding the morphology of undecalcified bone samples *in toto*. However, for two reasons, we decided to focus on paraffin samples in the presented study. First, we are confident that it is not reasonably possible to represent both groups of methods at the highest quality level in one research paper. Second, the evaluation of paraffin samples provides a unique opportunity for a variety of different stains, particularly focusing on the osteoimmunological response. For an in-depth discussion of resin sample analyses, we therefore refer the interested reader to our previous publications [20,25,32]. Indeed, to the best of our knowledge, our animal model is the first large-volume animal model of segmental tibial defects in sheep that comprehensively allows extensive and highly standardized analyses of tissue sections for short- and long-term outcomes via immunohistochemical staining.

In contrast to resin embedding, larger samples, such as those from large (3 cm) to extra-large (6 cm) ovine tibial segmental defects, pose technical challenges for paraffin embedding and further processing of paraffin samples. When standard cassette and block sizes (~25–40 mm) are limiting, oversized cassettes, embedding molds, and slides (~50–70 mm) can be used to preserve the integrity across large areas or minimize site fragmentation, but performing paraffin microtomy on *in toto* extra-large samples is not possible. Moreover, tissue sectioning is a critical step, and the choice of sectioning planes imparts an experimental bias that must be carefully interpreted. Notably, multiple and compound sectioning planes from various regions of interest may be necessary to thoroughly assess the tissue response. Therefore, in order to achieve meaningful ex vivo histological assessment and, in particular, analyze all anatomical regions in a consistent manner, we defined standardized transverse sectioning planes for the histological and immunohistological specimens and validated them for the large animal model (Appendix A).

This study aimed at profiling the adaptative osteoimmunological responses of our benchmark sheep research model as a pre-clinical tool for evaluating 3D-printed mPCL scaffolds applied to the reconstruction of critical-sized segmental bone defects, as opposed to making direct comparisons between the studied groups or time points. With regard to the time points analyzed, there are limitations, as although the samples were assessed at earlier and later time points, we are aware that the most important changes in the cellular immune response occur within the first few weeks after surgery [36]. It is important to mention that the testing time points were selected based on the utmost beneficial gain of data in regard to the 6Rs principle [76,77]. This can be corroborated via the circumstance that non-histological analysis, such as mechanical testing or µCT imaging, does not provide sufficient information at early implantation time points. However, differences can still be noted at chosen time points regarding immune cell phenotypes (M1 versus M2) and intercellular tissue quality (COL I versus COL II), which are to be interpreted as an evolving ongoing process of regeneration. Thus, a more accurate testimony can be made through quantification of the IHC data [26,78], which was not the focus of this paper for the reason that emphasis was given on providing an overview on immunohistochemical staining as a method for bone regeneration analysis, rather than comparing the outcome of the different study groups. However, IHC quantification was performed in our previous studies, and quantitative analytical methods were published and can be used in this context [78,79].

## 5. Conclusions

To steer regeneration, it is essential to understand how the immune response, upon mPCL-TCP scaffold implantation, degradation, and resorption, alters endogenous regenerative cascades over the entire period (>4 years) of regeneration. At present, it is thought that an increase or decrease in the immune response is primarily related to the biomechanical and physicochemical properties of the scaffold, as well as the baseline response modulated by the immune system, also defined as homeostasis. However, it is decisive that the anatomical location and extent of surgical bone defect creation, which strongly determine the active microenvironment of the peri-implant tissue [37], must also be considered as an immunologically active factor, as this shapes the activity of immunological sentinels, such as macrophages, to activate and control an immune-mediated and controlled inflammatory response. Notably, repeated surgical procedures, and the minimal surrounding muscle and soft tissue coverage, can lead to the disruption of the vasculature, resulting in an imbalance in the microenvironment that can contribute to a chronic imbalanced inflammatory response, and ultimately to failed implant integration. In contrast to the classic bone regeneration response, in which the activation state of macrophages (M1–M2 polarization) appears to play a critical role, this osteoimmune response, as part of a new model’s osteointegration, has been termed foreign body equilibrium [38]. In accordance with J. E. Davies [80], we observed variable activation states on a continuum between the M1 macrophages and M2 macrophages, which is in line with recent findings, indicating that macrophages do retain the ability to continue changing in response to new environmental stimulations [81]. Our data suggest that osteointegration and bone regeneration are osteoimmune responses rather than bone repair processes. State-of-the-art histological and IHC studies of SGBR therefore require protocols and assessment standards aimed at ensuring a fully comprehensive mapping of osteoimmunological spatiotemporal processes. We presented corresponding results related to IHC, focusing on the evaluation of specimens that consisted of SGBR studies in a well-characterized and validated segmental tibial sheep defect model.

## Figures and Tables

**Figure 1 biomedicines-11-02781-f001:**
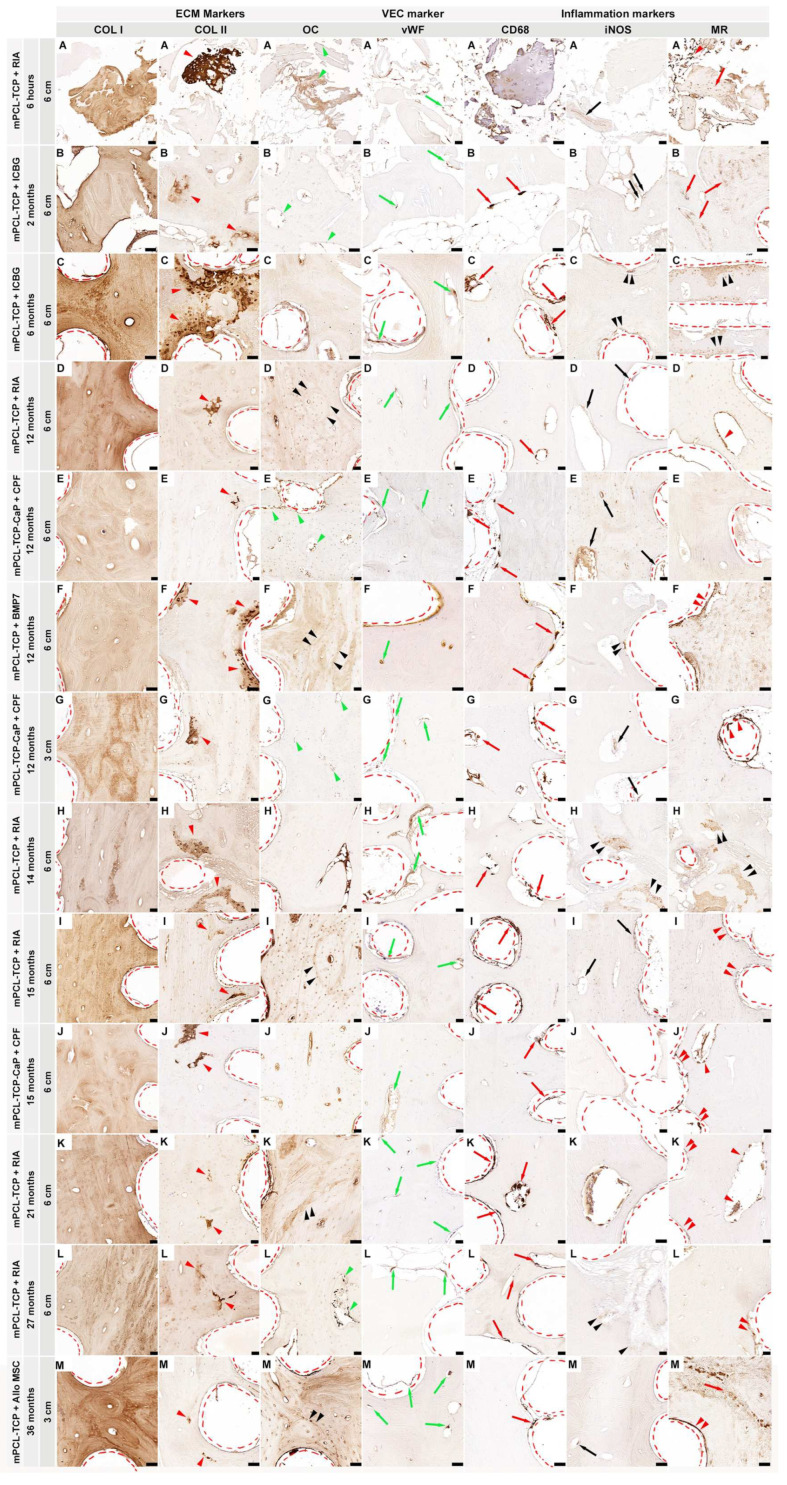
Immunohistochemical analysis of protein expression in SGBR throughout the entire bone regeneration period. Foreign-implanted materials, including mPCL scaffolds, activate an inflammatory response composed of a milieu of multiple converging cytokinetic and immunological events. This includes the provision of a collagen extracellular matrix (ECM) depicted through collagen type I (COL I, **A**–**M**) deposition, which is observed in all sample groups and during all time points. Clear differences in COL I stain intensity can be seen in woven bone (stronger) and lamellar bone (lighter). As bone formation in large bone defects is accomplished through osteochondral bone formation, it is common to observe the presence of a few remnant areas of mineralized cartilage going through resorption, as depicted by COL II deposition near the mPCL scaffold struts, as well as at the interface of host and new bone (COL II, **A**–**M**, red arrowheads). Concomitantly, the up-regulation of osteocalcin (OC) is observed at osteoblast precursors within the vascular mesenchymal tissue of the cutting cones of the remodeling bone tissue (green arrowheads) throughout all time points; however, with some distinct expression at the later time points, which appears to be more prominent at the cement lines of the new secondary osteons formed (black arrowheads) and at osteocyte cells (OC, **A**–**M**). As regeneration progresses, neovascularization also continues to re-establish the new vascular network, which is observed by the presence of more mature and elongated vessels through vWF+ expression throughout all time points and all experimental samples; however, this expression decreases around the outer surface of the mPCL-TCP scaffold towards the late time points (vWF, **A**–**M**, green arrows). As part of the innate immune system, early physiological wound healing is triggered by cytokines and cellular mediators, including macrophages. Cluster of differentiation 68 (CD68), as a M1 and M2 macrophage subset, was constantly positively stained around the interconnected porous architecture of the mPCL-TCP scaffold (CD68, **A**–**M**, red arrows). The pro-inflammatory inducible nitric oxide (iNOS–M1) synthesis observed in our studies appears to be evolving overtime, yet its expression was found to be located at the remnant areas of mineralized cartilage preceding blood vessel invasion at the early time points, and exclusively prevalent at blood vessels at the later time points (iNOS, **A**–**M**, black arrowheads and black arrows, respectively). Ultimately, M2 macrophages, depicted by the mannose receptor (MR) marker, appear to self-enhance their recruitment and play a crucial role within the host material’s consolidation and degradation, as observed throughout the expression and positive stain, particularly at the remnants of mineralized cartilage (MR, **A**–**M**, double black arrowheads), fragments of the bone graft (MR, **A**–**M**, red arrows), osteoblast precursors within the vascular mesenchymal tissue of the cutting cones (MR, **A**–**M**, red arrowheads), and at the outer surface of the mPCL-TCP scaffold struts (MR, **A**–**M**, double red arrowheads). This initial endogenous regenerative cascade, with further emerging evidence suggesting that the scaffold architecture is a niche where adaptative immune cells are decoding scaffold features, takes influence on the biocompatibility of materials, and appears to further orchestrate phenotypic macrophagic polarization by the presence of the CaP coating. Collagen type I (Col I); collagen type II (Coll II); osteocalcin (OC); cluster of differentiation 68 (CD68); inducible nitric oxidase synthesis (iNOS); and mannose receptor (MR). Red dashed lines: scaffold struts. Scale bars: 50 µm.

## Data Availability

Not applicable.

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
