# Peer review of "Histological and Immunohistochemical Characterization of Osteoimmunological Processes in Scaffold-Guided Bone Regeneration in an Ovine Large Segmental Defect Model"

_biomedicines, 2023, doi:10.3390/biomedicines11102781_

Round 1
Reviewer 1 Report
This study is interesting, it has an appropriate and detailed design that allows the replication of the study and the results obtained reflect the applied methodology supported by the statistical analysis of easy interpretation.The description of the figures is too long but understandable.
This manuscript presents the limitations of the study and the conclusions are appropriate to the results.
Author Response
The authors would like to thank the reviewer taking the time to review our manuscript.
Reviewer’s queries are in black colour font and authors responses are in red colour font. All changes made within the manuscript have been highlighted in yellow throughout the manuscript text.
Reviewer 1.
This study is interesting, it has an appropriate and detailed design that allows the replication of the study and the results obtained reflect the applied methodology supported by the statistical analysis of easy interpretation. The description of the figures is too long but understandable.
This manuscript presents the limitations of the study and the conclusions are appropriate to the results.
We thank the reviewer for taking the time to review our manuscript and for the positive feedback.
Reviewer 2 Report
Dear Authors,
I have read your paper "Histological and Immunohistochemical Characterization of Osteoimmunological Processes in Scaffold-Guided Bone Regeneration in an Ovine Large Segmental Defect Model" carefully.
The paper is easy to read.
Methods are properly described, so that other research groups may reproduce them.
The paper is interesting and useful for regenerative medicine.
The paper can be accepted for publication.
Author Response
The authors would like to thank the reviewer for taking the time to review our manuscript.
Reviewer’s comments are in black colour font and authors responses are in red colour font.
Reviewer 2.
Dear Authors,
I have read your paper "Histological and Immunohistochemical Characterization of Osteoimmunological Processes in Scaffold-Guided Bone Regeneration in an Ovine Large Segmental Defect Model" carefully.
The paper is easy to read.
Methods are properly described, so that other research groups may reproduce them.
The paper is interesting and useful for regenerative medicine.
The paper can be accepted for publication.
We thank the reviewer for taking the time to review our manuscript and for the positive feedback.
Reviewer 3 Report
This very interesting study delves into the intricate process of regenerating large volume bone defects, which is both time-consuming and multifaceted. The phases of regeneration, marked by distinct immune responses, are interconnected and must succeed to restore the bone's form and function. In clinical cases involving trauma, infection, or neoplasms, the natural bone regeneration capacity may fall short, necessitating surgical intervention. Scaffold-guided bone regeneration (SGBR) has shown promise in both preclinical and clinical contexts. To explore various SGBR strategies over a span of up to 3 years, we established a well-defined ovine model with large tibial bone defects and optimized immunohistochemistry (IHC) protocols. Our overview presents the IHC characterization of diverse experimental groups, all involving the treatment of ovine segmental defects with a bone grafting technique in conjunction with a three-dimensionally printed medical-grade polycaprolactone-tricalcium phosphate (mPCL-TCP) scaffold. Drawing upon qualitative data derived from over >350 sheep surgeries spanning two decades, our systematic and standardized IHC protocols have provided deeper insights into the intricate and protracted bone regeneration processes. This understanding proves pivotal for successful translational research in this field.
This study presents intriguing and innovative findings that warrant attention.
Firstly, I recommend that the authors discuss and if possible delve into the analysis and incorporation of data regarding type H endothelial cells PMID: 34281770 . These cells are well-established drivers of angiogenesis, and their inclusion in the discussion could provide valuable insights.
Additionally, it is crucial for the authors to discuss, explore and analyze the role of lymphatic vessels in bone regeneration PMID: 36669473. This can be achieved through techniques such as immunostaining or qPCR, as recent research has demonstrated their significant impact on the bone regeneration process. This addition would further enrich the study and enhance its overall impact.
This very interesting study delves into the intricate process of regenerating large volume bone defects, which is both time-consuming and multifaceted. The phases of regeneration, marked by distinct immune responses, are interconnected and must succeed to restore the bone's form and function. In clinical cases involving trauma, infection, or neoplasms, the natural bone regeneration capacity may fall short, necessitating surgical intervention. Scaffold-guided bone regeneration (SGBR) has shown promise in both preclinical and clinical contexts. To explore various SGBR strategies over a span of up to 3 years, we established a well-defined ovine model with large tibial bone defects and optimized immunohistochemistry (IHC) protocols. Our overview presents the IHC characterization of diverse experimental groups, all involving the treatment of ovine segmental defects with a bone grafting technique in conjunction with a three-dimensionally printed medical-grade polycaprolactone-tricalcium phosphate (mPCL-TCP) scaffold. Drawing upon qualitative data derived from over >350 sheep surgeries spanning two decades, our systematic and standardized IHC protocols have provided deeper insights into the intricate and protracted bone regeneration processes. This understanding proves pivotal for successful translational research in this field.
This study presents intriguing and innovative findings that warrant attention.
Firstly, I recommend that the authors discuss and if possible delve into the analysis and incorporation of data regarding type H endothelial cells PMID: 34281770 . These cells are well-established drivers of angiogenesis, and their inclusion in the discussion could provide valuable insights.
Additionally, it is crucial for the authors to discuss, explore and analyze the role of lymphatic vessels in bone regeneration PMID: 36669473. This can be achieved through techniques such as immunostaining or qPCR, as recent research has demonstrated their significant impact on the bone regeneration process. This addition would further enrich the study and enhance its overall impact.
Author Response
"Please see the attachment."

Round 2
Reviewer 3 Report
Authors have addressed my comments and I have no further comments
N/A